# An MMIC LNA for Millimeter-Wave Radar and 5G Applications with GaN-on-SiC Technology

**DOI:** 10.3390/s23146611

**Published:** 2023-07-22

**Authors:** Chaoyu Huang, Zhihao Zhang, Xinjie Wang, Hailiang Liu, Gary Zhang

**Affiliations:** 1School of Integrated Circuit, Guangdong University of Technology, Guangzhou 510006, China; 2112103147@mail2.gdut.edu.cn (C.H.); 2112103035@mail2.gdut.edu.cn (X.W.); garyzhang@gdut.edu.cn (G.Z.); 2School of Information Engineering, Guangdong University of Technology, Guangzhou 510006, China; 2112003031@mail2.gdut.edu.cn

**Keywords:** monolithic microwave integrated circuit (MMIC), 5G, GaN-on-SiC, low noise amplifier (LNA), GaN HEMT, stability handle, n257, n258, hybrid-MNs, near distance senor (NDS)

## Abstract

This paper presents a monolithic microwave integrated circuit (MMIC) low noise amplifier (LNA) that is compatible with n257 (26.5–29.5 GHz) and n258 (24.25–27.5 GHz) frequency bands for fifth-generation mobile communications system (5G) and millimeter-wave radar. The total circuit size of the LNA is 2.5 × 1.5 mm2. To guarantee a trade-off between noise figure (NF) and small signal gain, the transmission lines are connected to the source of gallium nitride (GaN)-on-SiC high electron mobility transistors (HEMT) by analyzing the nonlinear small signal equivalent circuit. A series of stability enhancement measures including source degeneration, an RC series network, and RF choke are put forward to enhance the stability of designed LNA. The designed GaN-based MMIC LNA adopts hybrid-matching networks (MNs) with co-design strategy to realize low NF and broadband characteristics across 5G n257 and n258 frequency band. Due to the different priorities of these hybrid-MNs, distinguished design strategies are employed to benefit small signal gain, input-output return loss, and NF performance. In order to meet the testing conditions of MMIC, an impeccable system for measuring small has been built to ensure the accuracy of the measured results. According to the measured results for small signal, the three-stage MMIC LNA has a linear gain of 18.2–20.3 dB and an NF of 2.5–3.1 dB with an input–output return loss better than 10 dB in the whole n257 and n258 frequency bands.

## 1. Introduction

Every decade, mobile communication takes a step forward, impacting not only people’s lifestyles but also driving the development and innovation of other industries. Based on current trends, countries with better 5G technology and research achievements will have more influence and dominance when the International Telecommunication Union (ITU) sets new standards, directly affecting a country’s strategic development and economic regulation in the future. The 3rd Generation Partnership Project defines three major application scenarios for 5G: Ultra-Reliable and Low Latency Communications (URLLC), Massive Machine Type Communication (mMTC), and Enhanced Mobile Broadband (eMBB). To prevent interference between different industries that use spectrum resources, the International Telecommunication Union has established international radio regulations. Currently, low-frequency (600 MHz–3 GHz) resources are mainly allocated from the first-generation mobile communications system (1G) to the fourth-generation mobile communications system (4G) [1,2]. The most significant change in the communication generation from 2G to 5G is the increase in data transmission speeds. According to Shannon’s law, the main strategies to enhance data transmission speeds are to increase the bandwidth, increase the number of receive/transmit channels, and improve the Signal-to-Noise Ratio (SNR). By moving up to millimeter-wave (mmWave), a high data rate, wide channel bandwidth, and low latency can be achieved simultaneously [3,4]. In addition, the 5G mmWave frequency bands n257 and n258 may receive global prioritization. Such regulations are distinguished from the lower, overcrowded sub-6 G frequency range and cater for the requirements of high data rate wireless communications in today’s world.

Millimeter-wave radar systems [5,6] are usually divided into Frequency Modulated Continuous Wave (FMCW) radar and Doppler radar. Considering the relatively higher maximum equivalent isotropic radiated power (EIRP) allowed in the standard and the relatively narrower bandwidth requirement compared with the Doppler radar, FMCW radar is more attractive for millimeter-wave radar applications. Based on the FMCW radar system, the LNA is one of the most pivotal active components, which highly determines the overall NF of the receiver. To meet the requirements of dynamic range and sensitivity of the millimeter-wave radar system, we need to design an MMIC LNA with low NF, reasonable gain, and broadband bandwidth, which further introduces great challenges in finding a trade-off between these performances [7]. 

Compared with gallium arsenide (GaAs), GaN-based materials can effectively reduce the number of transceivers and circuit size while providing higher power handling ability and higher cutoff frequency [8,9,10,11]. Due to the high expenditure of using homogeneous materials, heterogeneous materials are mostly used for GaN technology, including GaN-on-Si and GaN-on-SiC. Since the thermal conductivity of GaN-on-SiC is three times higher than that of GaN-on-Si, devices can operate at higher voltages and higher power densities. GaN-on-SiC HEMT devices have become one of the best candidates for mmWave power amplifiers (PAs) owing to their distinct benefits [12,13,14,15]. Although many studies primarily focus on analyzing the power performance of GaN HEMT, its noise performance [16,17,18,19] has also attracted increasing attention. In fact, GaN HEMTs achieve considerable microwave noise performance comparable to gallium arsenide (GaAs). Additionally, the excellent robustness of GaN transistors and the ability to integrate PAs and LNAs in the same epitaxial material without the need for receiver protection circuits have sparked great interest in GaN technology for low-noise applications. The RF front-end module, which integrates LNA, PA, and switch, has lower package loss and is more cost-effective, resulting in reduced size.

A plethora of existing literature has documented the development of LNAs exhibiting satisfactory performance across various technologies [20,21,22,23,24,25,26,27,28,29]. As for the CMOS process, scholars of City University of Hong Kong [20] have presented an MMIC LNA which achieves a peak gain of 10.2 dB with a minimum NF of 3.3 dB and bandwidth ranging from 15.8 GHz to 30.3 GHz. Furthermore, the National University of Singapore [24] has put forth a groundbreaking proposal for an ultra-wide bandwidth cascode LNA, utilizing GaAs pHEMT technology. This innovative design boasts a peak gain of 14.1 dB and a notable NF below 4.5 dB, effectively covering an extensive frequency range from 2 to 42 GHz. Ho Chi Minh City University of Technology [25] has presented a 35–37 GHz Ka-band MMIC LNA with a typical NF of 3.7 dB and a small signal gain of above 19.7 dB employing GaN-on-SiC HEMT. A K-band MMIC LNA [26] in GaN-on-Si technology has realized a measured average gain of 17.2 dB and an NF of 2.0–2.3 dB. These advancements pave the way for further research and innovation, enabling the design and implementation of LNAs tailored to specific application requirements and operating conditions.

In this letter, we realize a cascaded common-source three-stages [30] MMIC LNA covering 24.25–29.5 GHz with a minimum NF of 2.5 dB and a maximum small signal gain of 20.2 dB based on the 0.15 μm GaN-on-SiC HEMT process developed by WIN foundry. To enhance the stability of a designed LNA, a series of stability enhancement measures including source degeneration, RC series network, and RF choke [30,31] are put forward. The hybrid-matching networks (MNs) with a co-design strategy are adopted to realize low NF and broadband performance. Due to the different priorities of these hybrid-MNs, distinguished design strategies are employed. In order to meet the testing conditions of the MMIC, an impeccable system for measuring small signals has been built to ensure the accuracy of the measured results.

## 2. Analysis and Design of MMIC LNA

In the field of low NF circuit design, the selection of an optimal topology for the GaN HEMT LNA holds significant importance. The effectiveness and performance of the LNA greatly depend on the chosen topology. In this particular study, the MMIC LNA’s topology is explicitly shown in Figure 1, providing a visual representation of its design. In order to fulfill the requirement for small signal gain, the MMIC LNA is built with an architecture consisting of a three-stage common-source (CS) amplifier [30]. By utilizing a multi-stage approach, the LNA can achieve higher gain levels while maintaining low noise performance. One critical factor that needs to be taken into account when designing the LNA is the Miller effect caused by the gate-to-drain capacitance (C_gd_). To counteract this effect and minimize its impact on amplifier performance, an RC series network is introduced. This network is connected in parallel to the drain of the GaN HEMT, accompanied by a series λ/4 microstrip line. This combination effectively reduces the undesirable consequences of C_gd_, thus enabling better overall performance of the LNA. Another aspect to consider is the detrimental impact of excessive input power on the LNA. When the input power exceeds certain limits, it can lead to a direct current (dc) harmonic surpassing the breakdown current. To mitigate this potential issue, a resistor with a kilo-ohm range has been integrated into the gate bias feed. By incorporating this resistor, the DC gate current can be limited effectively, providing additional protection to the LNA, and will help against the high-input-power-induced failures. Additionally, the equivalent model of the MNs is also depicted in the illustrated topology. These networks play a crucial role in ensuring appropriate impedance matching between different components of the LNA, optimizing power transfer, and maximizing efficiency. A comprehensive explanation of the various structures incorporated within the topology will be provided in subsequent sections, offering a detailed understanding and analysis of the design choices made in this study.

### 2.1. Device Characteristics

The MMIC LNA was meticulously designed utilizing coplanar waveguide (CPW) transistors within the 0.15 μm GaN-on-SiC process, provided by WIN foundry. The initial version of the CPW transistor utilized [S] parameter file, which imposed certain limitations on the available choices for bias voltage and current [32]. It should be noted that reducing the gate width of the CPW transistor has the potential to lower the minimum noise figure (NFmin). However, this reduction may also result in a compromised bandwidth of the LNA. Hence, it becomes imperative to strike a delicate balance between NF and bandwidth, taking into account the specific design requirements and objectives. To address this tradeoff, the three-stage amplifier in this design adopted the use of a 2 × 50 μm GaN HEMT, which has a breakdown voltage exceeding 120 V, a current gain cutoff frequency (f_T_) of approximately 34.5 GHz, and a maximum oscillation frequency (f_MAX_) of around 125 GHz. Additionally, this process provides two interconnected metal layers, high-reliability-metal-insulator-metal capacitors, and precision tantalum nitride (TaN) resistors for comprehensive MMIC designs. On one hand, the active intrinsic core of the GaN HEMT lends itself to being ideally modeled as noiseless two-ports housing correlated noise sources at both the input and output. As for the modeling of nonlinear parasitic effects associated with the GaN HEMT transistor, an intricate network comprising lumped elements was employed. Leveraging the well-established theory of inductive source degeneration, the CPW transistor’s source was connected to an inductor. To make the pre-appraisal of the GaN HEMT noise characteristic more accurate, a simplified representation of the GaN HEMT’s noise equivalent circuit model [16,17,18,19] is presented in Figure 2, incorporating a source series inductor [1,12,33]. This model only considers the shot noise and channel diffusion noise since the channel diffusion noise accounts for a large proportion of device noise.

This comprehensive design approach ensures that both the active and passive elements of the MMIC LNA are appropriately characterized and optimized for performance. By carefully considering the tradeoffs and leveraging established theoretical frameworks, researchers can make informed decisions to achieve desired outcomes in terms of noise figure, bandwidth, and overall performance characteristics.

The impact of R_d_ on noise can be ignored compared to the thermal noise of the channel, as R_d_ is at the output node (drain) of the GaN HEMT. Since we only applied for a rough estimate of the noise impedance, the effects of parasitic parameters such as inductance of the source and inductance of the gate are ignored. The Z_opt_ (optimum noise impedance) of the circuit is found to be as follows:
(1)$$Z_{\mathrm{opt}}=\frac{\sqrt{g_{m}(R_{S}+R_{G})+\frac{PR(1-C^{2})}{P+R-2C\sqrt{PR}}}}{P+R-2C\sqrt{PR}}\frac{1}{wC_{gs}}+j\frac{1}{wC_{gs}}\frac{P-R\sqrt{PR}}{P+R-2C\sqrt{PR}}$$

P and R are the factors jointly determined by both the internal and external elements of the model [16,17,18,19]. Since the noise current source is not independent, C is the noise correlation coefficient of I_g_ and I_d_. As shown in Equation (1), we can optimize the Z_opt_ by modulating the C_gs_ which can be adjusted by changing the gate width of transistor.
(2)$$\begin{array}{l} Z_{\mathrm{in}}=\frac{V_{in}}{I_{in}}=R_{i}+\frac{1}{jwC_{gs}}+jwL_{S}+\frac{g_{m}L_{S}R_{ds}}{C_{gs}.(R_{ds}+R_{i}+\frac{1}{jwC_{ds}}+jwL_{s})}\\ +\frac{{\left(wL_{S}\right)}^{2}}{R_{ds}+R_{i}+\frac{1}{jwC_{ds}}+jwL_{s}}+\frac{g_{m}L_{S}\frac{1}{jwC_{ds}}}{C_{gs}.(R_{ds}+R_{i}+\frac{1}{jwC_{ds}}+jwL_{s})} \end{array}$$

When $jwL_{s}+\frac{1}{jwC_{ds}}<<R_{ds}+R_{i}$, based on Equation (2), Z_in_ can be expressed as follows:
(3)$$Z_{\mathrm{in}}=R_{i}+\frac{g_{m}L_{S}R_{ds}+g_{m}L_{S}\frac{1}{jwC_{ds}}}{C_{gs}.(R_{ds}+R_{i})}+\frac{{\left(wL_{S}\right)}^{2}}{R_{ds}+R_{i}}-jw(\frac{1}{C_{gs}w^{2}}-L_{s})$$
(4)Re[Zin]=Ri+gmLSRdsCgs.(Rds+Ri)+(wLS)2Rds+Ri
(5)Im[Zin]=gmLS1jwCdsCgs.(Rds+Ri)−jw(1Cgsw2−Ls)

From Equations (4) and (5), with the increase of L_S_, the real part of Z_in_ becomes lager, while the imaginary part of Z_in_ is slightly affected.

A thorough analysis of the simulation data reveals that tuning L_S_ enables engineers to optimize the impedance characteristics of the design, effectively tailoring it to meet specific requirements. The ability to manipulate L_S_ allows for fine-tuning the input impedance, ensuring optimal matching with the external circuitry or transmission line. This capability is crucial in guaranteeing efficient power transfer, minimizing reflections, and maximizing overall performance. Furthermore, the widening and flattening of the bandwidth contribute to enhanced signal integrity and improved overall system performance. A wider bandwidth allows for greater frequency coverage, enabling the LNA to operate effectively across a broader range of frequencies. This is particularly advantageous in applications where the input signal may vary in frequency or when dealing with multi-band communication systems. The observed relationship between L_S_, input impedance, and bandwidth underscores the importance of carefully considering the choice of L_S_ during the design phase. It serves as a parameter that can be adjusted to achieve the desired impedance characteristics and optimize the performance of the LNA in terms of both matching and frequency response. It is worth noting that these findings are based on the simulation results presented in Figure 3, which provide valuable insights into the behavior of the LNA under various L_S_ configurations. 

The LNA’s NF is influenced by both the NF_min_ and the source impedance (Z_S_). After identifying the minimum attainable NF, it becomes possible to ascertain the optimal Z_S_ (Z_opt_). Compared to NF_min_, amplifiers without matching circuits undoubtedly exhibit significantly poorer NF. However, we can adopt some effective measures to reduce this gap. Figure 4 illustrates that the NF of the GaN HEMT cell has improved by approximately 0.7 dB with the working frequency band. 

When designing the IMN for LNA, it is necessary to try to make the S11 curve intersect the curve of Voltage Standing Wave Ratio (VSWR) = 2.0 curve on the smith chart. As shown in Figure 5, introducing Ls causes the reflection coefficient to move towards the real axis and approach to the circle of VSWR = 2.0. By implementing the above measures, the values of the components used in the input matching circuit can be reduced, thereby minimizing the frequency offset caused by device errors.

Figure 6 depicts the values of forward gain S21 and reverse gain S12 for a GaN HEMT cell without MNs design. It can be observed that within the operating frequency range, the forward gain exhibits minor attenuation, while the performance of reverse gain has been significantly improved.

These results serve as a foundation for further refinement and optimization of the design, allowing engineers to make informed decisions and tradeoffs to achieve the desired performance objectives.

### 2.2. Stability Analysis and Enhancement

As depicted in Figure 7, a comprehensive and robust LNA stability handling topology is implemented for each stage of the amplifier. This topology incorporates various techniques to ensure optimal stability and performance throughout the LNA’s operation. One notable aspect of this design is the use of a microstrip line for source degeneration instead of an inductor, which introduces an additional stabilization mechanism. By employing a microstrip line for source degeneration, several advantages are gained. The use of a microstrip line for source degeneration offers greater flexibility and ease of integration compared to traditional inductors. These modifications contribute to cost savings and enable easier integration into compact and space-constrained electronic systems. This not only contributes to cost savings but also enables easier integration into compact and space-constrained electronic systems. The effectiveness of this stability handling topology is further validated through extensive simulations and measurements. As shown in Figure 8, the stable conditions for the LNA are met, ensuring reliable and predictable performance across the desired frequency range. However, it is important to note that while this topology enhances stability, additional measures are necessary to achieve unconditional stability (k > 1) throughout the entire operating frequency band. To achieve unconditional stability, designers employ various techniques such as adding feedback networks, optimizing biasing schemes, employing compensation techniques, and utilizing active or passive devices with appropriate impedance terminations. These measures work in concert to maintain stability across the LNA’s entire frequency range, preventing instability and ensuring reliable operation under different input signal conditions. By adopting the aforementioned stability handling topology and implementing additional stabilization measures, the LNA can meet stringent stability requirements while delivering high-performance characteristics across a wide frequency band. This comprehensive approach to stability design helps minimize the risk of instability, ensure reliable operation, and maximize the overall performance of the LNA in practical applications.

The π-type two-port network of the GaN HEMT [1,12] can be simplistically presented as Y-parameter matrix:(6)$${\left[Y\right]}_{HEMT}=\left[\begin{array}{cc} Y_{A}+Y_{B} & -Y_{B}\\ -Y_{B} & Y_{B}+Y_{C} \end{array}\right],$$

After connected in parallel with voltage-controlled current source, the intrinsic Y-parameter model can be written as follows:(7)$${\left[Y\right]}_{HEMT}=\left[\begin{array}{cc} \frac{jwC_{gs}}{jwC_{gs}R_{i}+1} & -jwC_{gd}\\ -jwC_{gd}+\frac{g_{m}}{jwC_{gs}R_{i}+1} & \frac{1}{R_{ds}}+jw(C_{ds}+C_{gd}) \end{array}\right],$$

The intrinsic $S_{12}$ can be written as follows:(8)$$S_{12}=\frac{-2Y_{12}Z_{0}}{\Delta }=\frac{2jwC_{gd}Z_{0}}{(1+Z_{0}Y_{11})(1+Z_{0}Y_{22})-Y_{12}Y_{21}Z_{0}^{2}},$$
where Z_0_ represents the characteristic impedance of the GaN HEMT. Based on Equation (8), the inverse gain S_12_ is mainly caused by C_gd_, which introduces a potential instability factor while concurrently restricting high-frequency performance.

As depicted in Figure 7, an RC series network is connected in parallel to the drain of the GaN HEMT, complemented by a series λ/4 microstrip line, to mitigate the Miller effect caused by C_gd_. This topology is combined with another capacitance to form the RF choke. The RF chokes determine the stability of each stage by blocking the RF signal to the bias network and decoupling the power supply. And then, the use of RF chokes effectively prevents the α signal path (as shown in Figure 7), further improving the stability of the circuit.

### 2.3. Co-Design Strategy for Broadband Matching and Low NF

As demonstrated in Section 2.2, the C_gd_ of the GaN HEMT cannot be overlooked in the mmWave band. This results in a rapid decrease in small signal gain, poor flatness, and inferior stability. In the design of mmWave LNA, matching networks have a significant impact on small signal gain, input-output return loss, and NF performance. Therefore, adopting appropriate topologies is crucial [34]. The simplified schematic of the proposed LNA is shown in Figure 1, which is based on a three-stage CS design.

In Figure 1, four hybrid-MNs are employed to fulfill the requirement of appropriate matching for broadband and low NF [1,4,26]. In order to minimize the introduction of noise, the design strategy for the input matching network (IMN) proposes the use of few lumped components [35].

As shown in Figure 9a, considering the high-frequency equivalent model and through a virtual resistor R_i1_, the IMN can be equivalent to three L-type MNs (L_0_C_L0_, L_1_C_01_, and C_02_L_λ1_) [1], where L_0_ and C_L0_ represent equivalent models of TL_0_, L_1_ represents an equivalent model of TL_1_, L_λ1_ is provided by the bias network, and $C_{0}=\frac{C_{01}C_{02}}{C_{01}+C_{02}}$ [31]. A virtual resistor R_i1_ is generated by using the impedance transformation network C_02_L_λ1_. The quality factor Q_3_ of C_02_L_λ1_ can be expressed as follows:(9)$$Q_{3}=w_{0}C_{02}R_{i1}=\sqrt{\frac{R_{L1}}{R_{i1}}-1},$$
where $R_{i1}=\frac{R_{L}}{1+Q_{3}^{2}}$. Through the same strategy, the quality factor Q_I_ of the whole hybrid-IMN can be indicated as follows:(10)$$Q_{I}=w_{0}(C_{01}+C_{02})R_{i1}+\frac{w_{0}L_{0}}{R_{S1}}=\sqrt{\frac{R_{L1}}{R_{i1}}-1}+\sqrt{\frac{R_{i1}}{R_{i2}}-1}+\sqrt{\frac{R_{i2}}{R_{s1}}-1},$$

The value of Q_I_ needs to be carefully designed by choosing the appropriate impedance. If the value of Q_I_ is too large, the small signal gain will decrease; conversely, the bandwidth of IMN will increase [26,33].

Through the same analysis method, the equivalent model of ISMNs is shown in Figure 9b. Considering the high-frequency equivalent model and introducing a virtual resistor R_i_, the IMN can be equivalent to two L-type MNs (L_2_C_L2_ and C_1_L_λ2_), where L_2_ and C_L2_ represent the equivalent model of TL_2_, and L_λ2_ is provided by the bias network. The quality factor Q_IS_ of the whole hybrid-ISMN can be indicated as follows:(11)$$Q_{IS}=w_{0}C_{1}R_{i}+\frac{w_{0}L_{2}}{R_{S2}}=\sqrt{\frac{R_{L}}{R_{i}}-1}+\sqrt{\frac{R_{i}}{R_{S2}}-1},$$

To mitigate these challenges, careful selection of high-impedance stages at both ends of the MMIC LNA can be employed. By strategically choosing these stages, the value of the quality factor Q_IS_ can be controlled effectively, which in turn helps to suppress the loss of performance caused by high-order harmonics and parasitic capacitance [26]. This approach enables designers to optimize the performance of the LNA by minimizing the negative impact of these unwanted effects.

Similar to the IMN, the hybrid OMN is designed to broaden the bandwidth of the MMIC LNA, ensuring that the amplifier operates efficiently across a wider range of frequencies. It is worth noting that the design of the OMN should be optimized in conjunction with other aspects of the MMIC LNA. These combined efforts help achieve a well-balanced and high-performing MMIC LNA that meets the specific requirements of the target application.

In summary, addressing issues related to high-order harmonics and parasitic capacitance is crucial in MMIC LNA design. Careful selection of high-impedance stages can control the value of Q_IS_ and suppress performance losses. The OMN and ISMNs are designed to widen the bandwidth and maintain inter-stage impedance matching. By considering these factors, designers can optimize the performance of the MMIC LNA, ensuring optimal small-signal gain, NF, and input/output return loss characteristics.

## 3. Implementation and Measurement

The micrograph of the fabricated three-stage LNA is shown in Figure 10, where the effective dimension marked in the red box is 2.5 × 1.5 mm^2^. The die is fabricated using 0.15 μm GaN-on-SiC technology and has a larger footprint than necessary. This is achieved entirely to match the die area of other MMIC modules in the multi-project. The GaN HEMT is supplied with a 5 V drain bias voltage through the VD, and its −1 V source bias voltage is supplied through the VG, respectively. RF signals enter from the RF_in_ port and exit from the RF_out_ port.

Figure 11 illustrates a set of equipment block diagrams for small signals, including S-parameters and NF [2,36,37]. A 5V supply voltage, provided in pulse mode by the RIGOL DP832A, is used for the drain bias. This adjusts the gate bias voltage to obtain the same quiescent current. The small signal performance of the MMIC LNA is measured using the Keysight PNA Vector Network Analyzer N547B, which operates within the n257–n258 frequency range. And the simulation results were performed by using ADS2020 software with a GaN HEMT design kit.

Figure 12 and Figure 13 present the measured and simulated S-parameter results of the proposed MMIC LNA from 20 GHz to 35 GHz. The measurements demonstrate that the implemented LNA exhibits a linear small signal gain (S21 refer to Figure 12) from 18.3 dB to 20.2 dB. Additionally, the reflection coefficients (S11 and S22, as shown in Figure 13) remain below less −10 dB across the frequency range of 24.25–29.5 GHz. The measured reverse isolation (S12, as shown in Figure 13) of the fabricated MMIC LNA is better than −40 dB. Figure 12 and Figure 13 indicate that the test results exhibit good agreement with the simulation results, albeit with a certain degree of degradation in both cases.

The measured NF response of the proposed LNA in Figure 14 is 2.5–3.1 dB in the interesting frequency band of 24.25–29.5 GHz, which is about 0.7 dB worse than the simulated result. It is customary for the test results of noise figure (NF) to exhibit degradation compared to simulation performance. This discrepancy can be attributed to the presence of unaccounted parasitic effects during the simulation process. 

As shown in Figure 15, measurements of k and b indicate that the LNA is stable over the entire frequency band, thanks to a series of strategies adopted in this design to improve stability.

Table 1 summarizes the performance comparison of some related LNAs with different processes, which demonstrates the advanced performance of this designed MMIC LNA. As can be seen from Table 1, although this design is not the best in terms of bandwidth, gain, noise factor, power consumption, and other performance metrics, it still exhibits advanced characteristics based on the aforementioned performance. In comparison to designs utilizing the same GaN process, this design not only achieves lower NF but also consumes less power. Moreover, when compared to designs using GaAs and CMOS processes, this design demonstrates significant advantages in terms of small signal gain and NF.

## 4. Conclusions

In this research paper, we present a comprehensive study on the design and implementation of a three-stage broadband MMIC LNA. The LNA operates in the frequency band of 24.25–29.5 GHz and is fabricated using an advanced 0.15 μm GaN-on-SiC HEMT process. To meet the requirements for gain and NF, an in-depth analysis of the device characteristics is conducted in Section 2.1. This analysis allows us to gain a deep understanding of the device behavior and select appropriate design strategies to optimize the LNA performance. Several stability enhancement measures are proposed in this paper, including the implementation of source degeneration, RC series networks, and RF chokes. These measures effectively improve the stability of the amplifier, preventing oscillations and ensuring reliable operation across the desired frequency range. To achieve a balance between low noise and wide bandwidth, hybrid-MNs with a co-design strategy are adopted. This approach combines different types of matching networks to maximize the benefits of each, resulting in improved overall performance. The proposed 3.75 mm^2^ MMIC LNA exhibits a measured small signal gain of 18.3–20.2 dB and a minimal NF of 2.5 dB in the 24.25–29.5 GHz frequency range. It offers excellent amplification capabilities while maintaining a low NF, making it well-suited for sensitive receiver applications in these fields. Through detailed analysis, innovative stability enhancement measures, and the adoption of a hybrid-MNs design strategy, we have successfully designed and implemented a broadband MMIC LNA that meets the demanding requirements of 5G and millimeter-wave radar applications. The experimental results validate the effectiveness of our approach, highlighting the potential for further advancements and applications in the field of integrated circuit design and RF engineering.

## Figures and Tables

**Figure 1 sensors-23-06611-f001:**
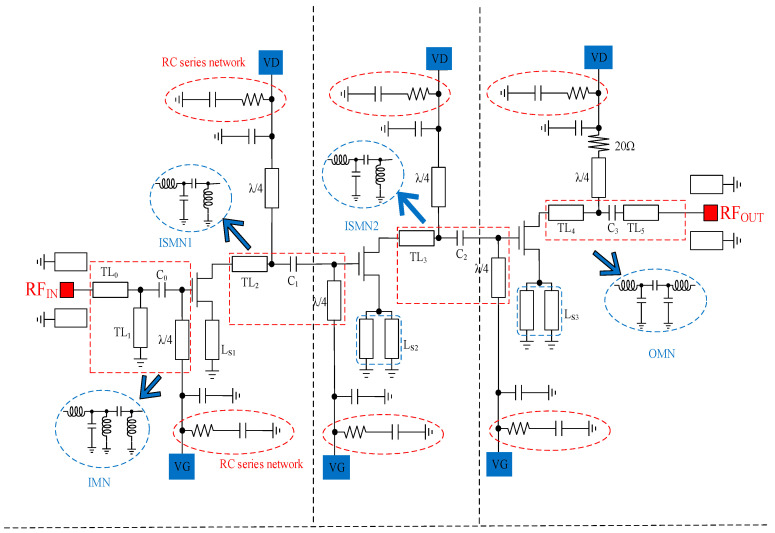
Simplified schematic of proposed LNA.

**Figure 2 sensors-23-06611-f002:**
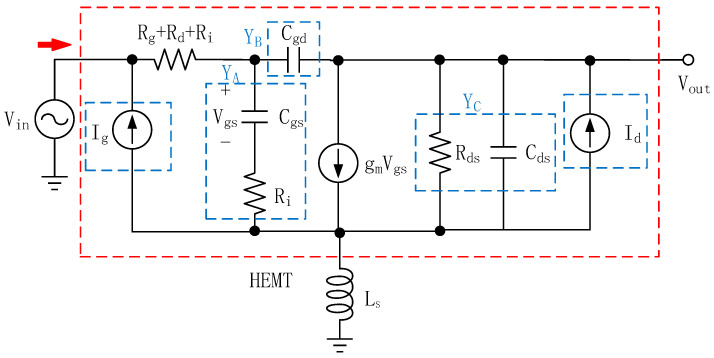
Noise equivalent circuit model of GaN HEMT with source series inductor.

**Figure 3 sensors-23-06611-f003:**
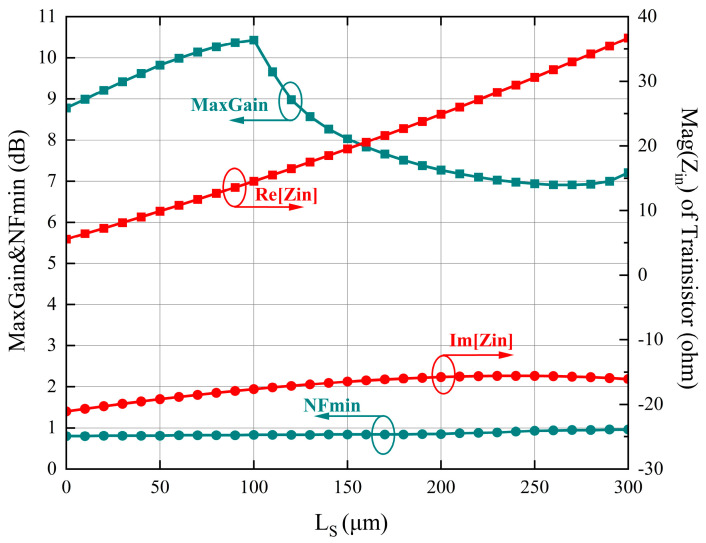
A 2 × 50 μm CPW transistor characteristics versus source L_S_.

**Figure 4 sensors-23-06611-f004:**
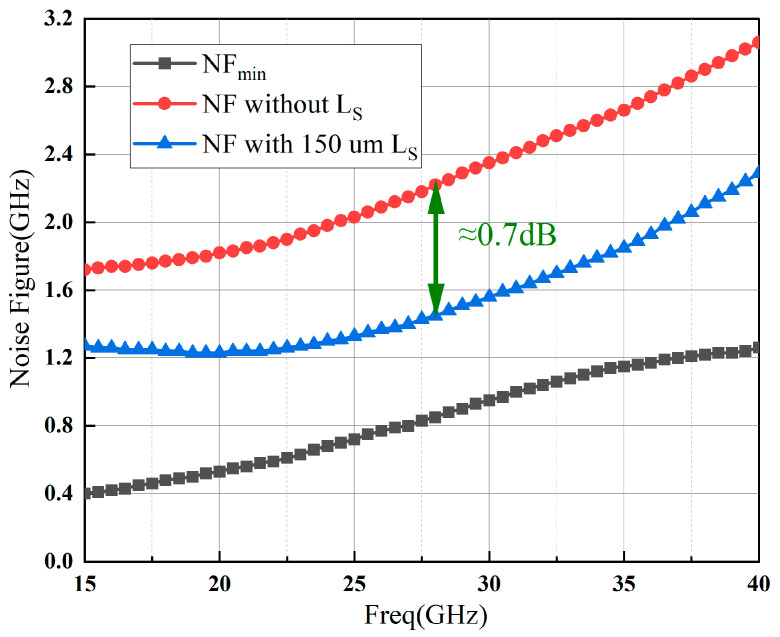
Simulated noise of GaN HEMT cell based on 50-ohm.

**Figure 5 sensors-23-06611-f005:**
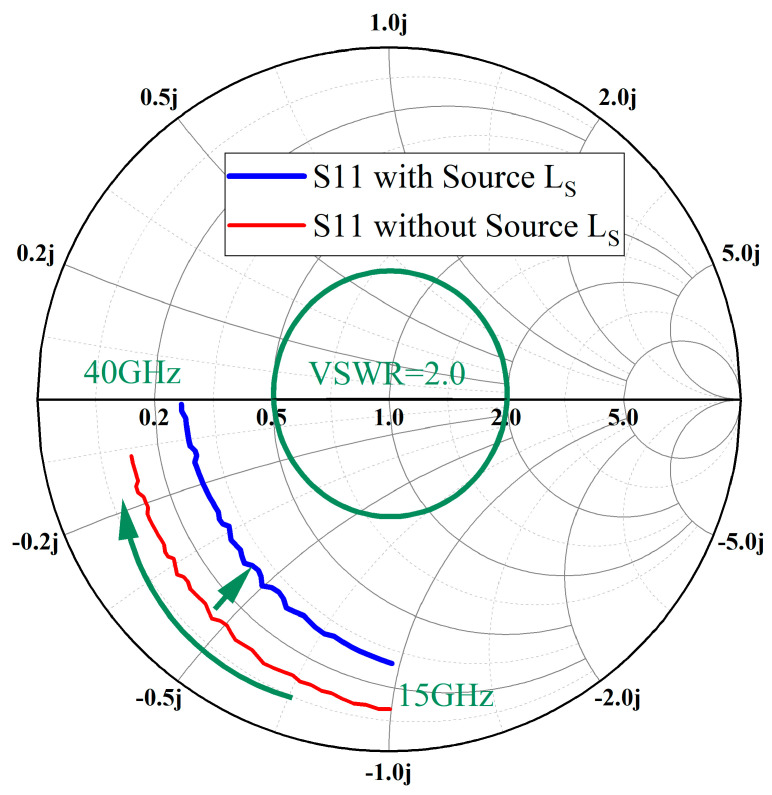
Simulated reflection coefficient of GaN HEMT cell based on 50-ohm.

**Figure 6 sensors-23-06611-f006:**
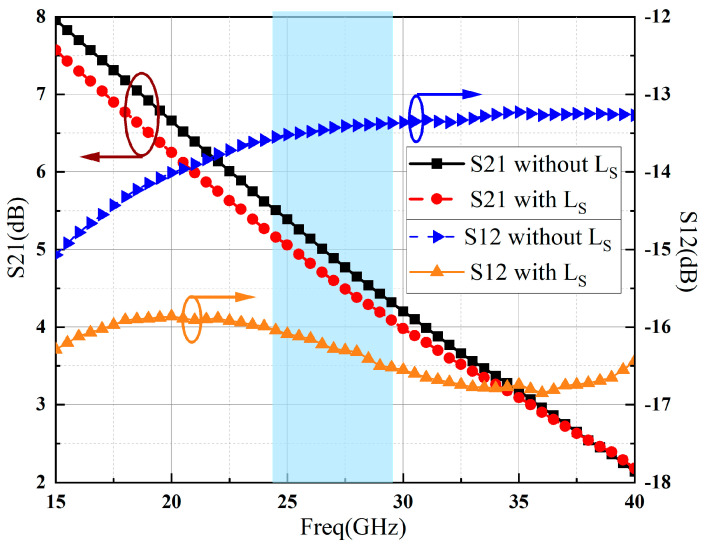
Simulated S-parameters of GaN HEMT cell based on 50-ohm.

**Figure 7 sensors-23-06611-f007:**
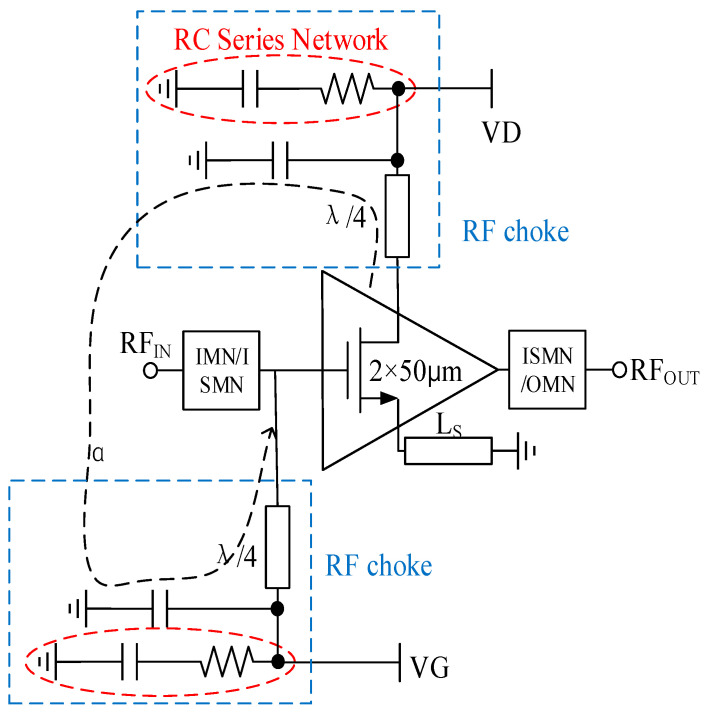
Stability handle topology.

**Figure 8 sensors-23-06611-f008:**
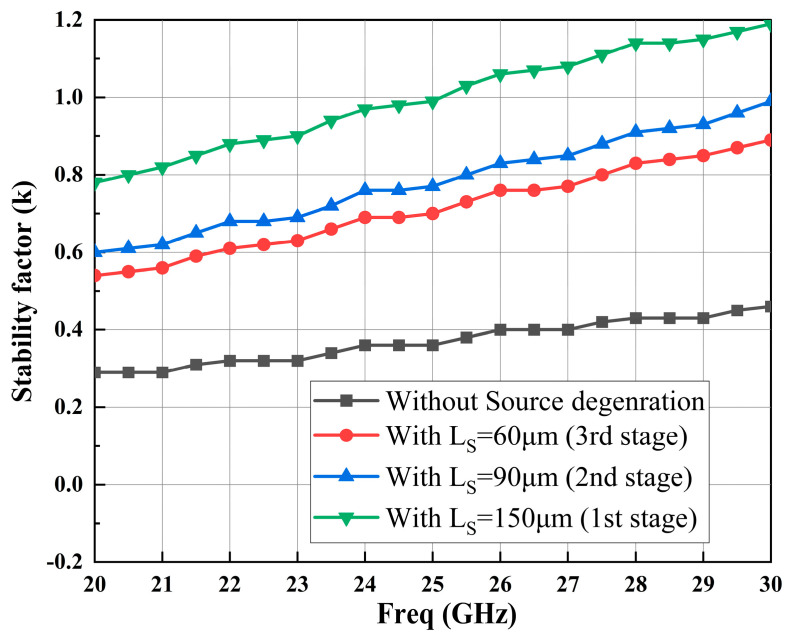
Source degeneration for stability factors of each stage.

**Figure 9 sensors-23-06611-f009:**
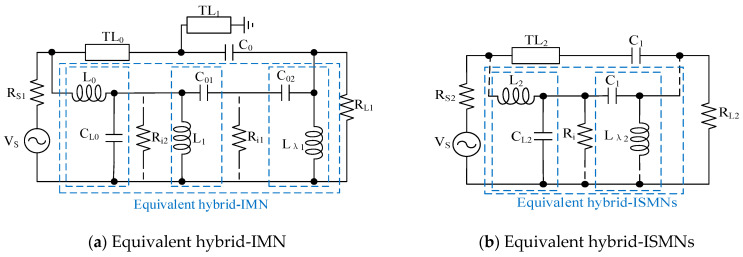
Hybrid matching network.

**Figure 10 sensors-23-06611-f010:**
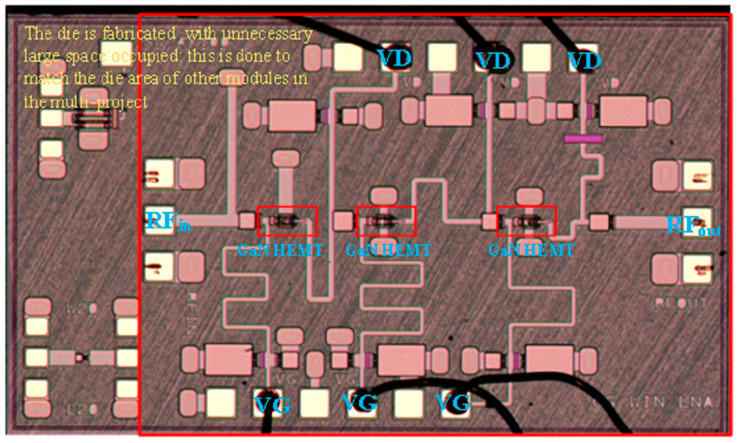
Micrograph of the proposed LNA (dimensions: 2.5 × 1.5 mm^2^).

**Figure 11 sensors-23-06611-f011:**
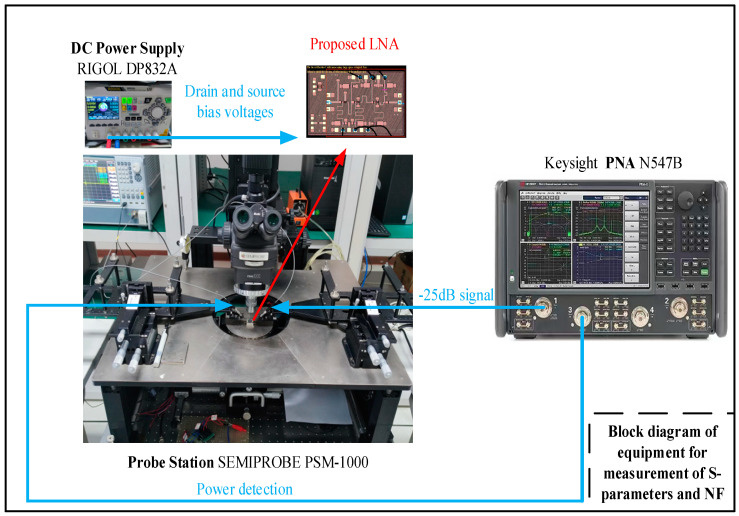
Block diagram of equipment for measurement of S-parameters and NF.

**Figure 12 sensors-23-06611-f012:**
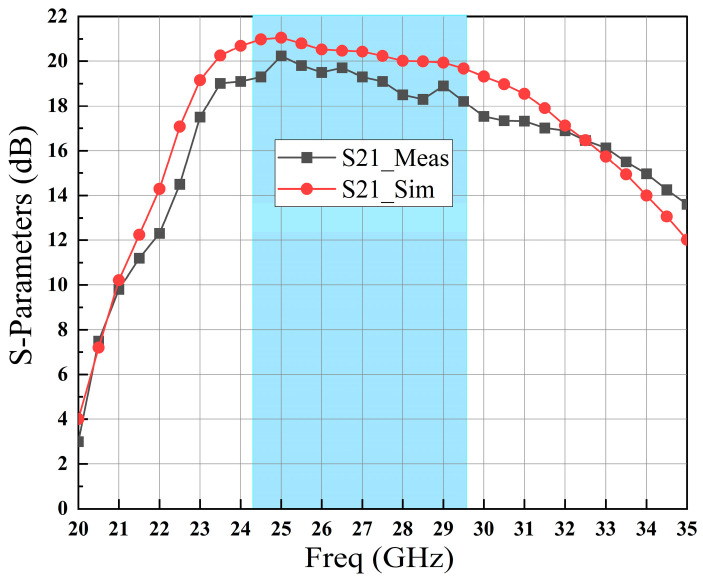
Measurement and simulation results of S21 for the fabricated MMIC LNA.

**Figure 13 sensors-23-06611-f013:**
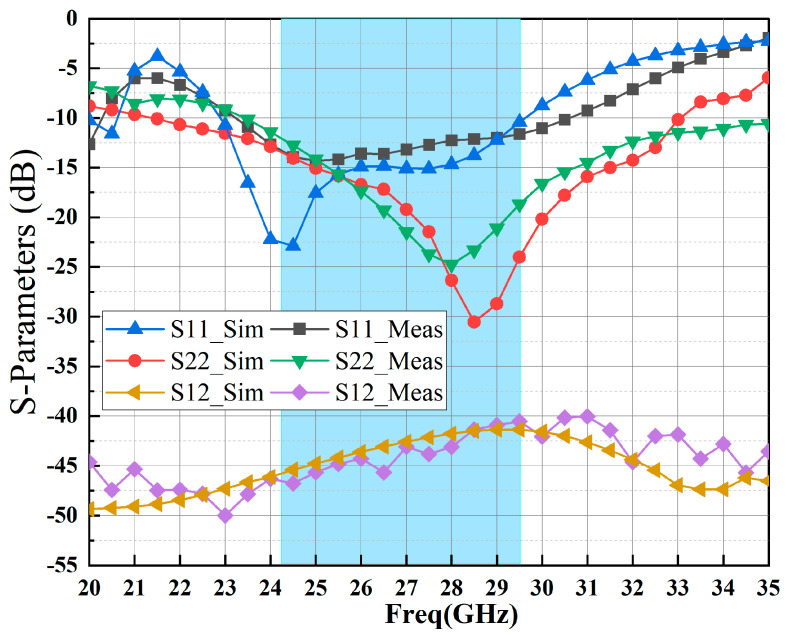
Measurement and simulation results of S-parameters (S11, S22, and S12) for the fabricated MMIC LNA.

**Figure 14 sensors-23-06611-f014:**
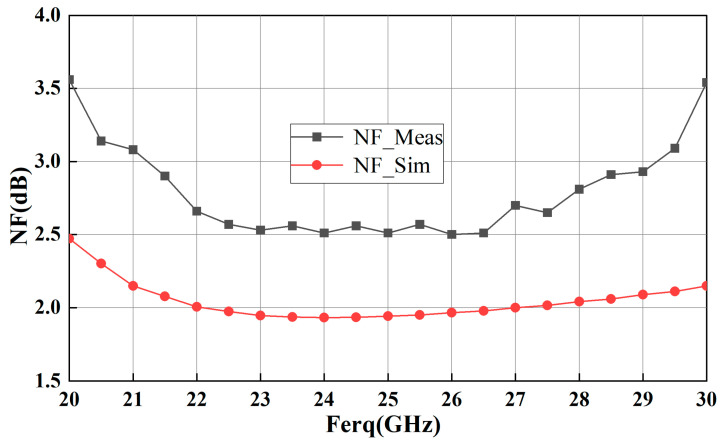
Measurement and simulation results of NF for the fabricated MMIC LNA.

**Figure 15 sensors-23-06611-f015:**
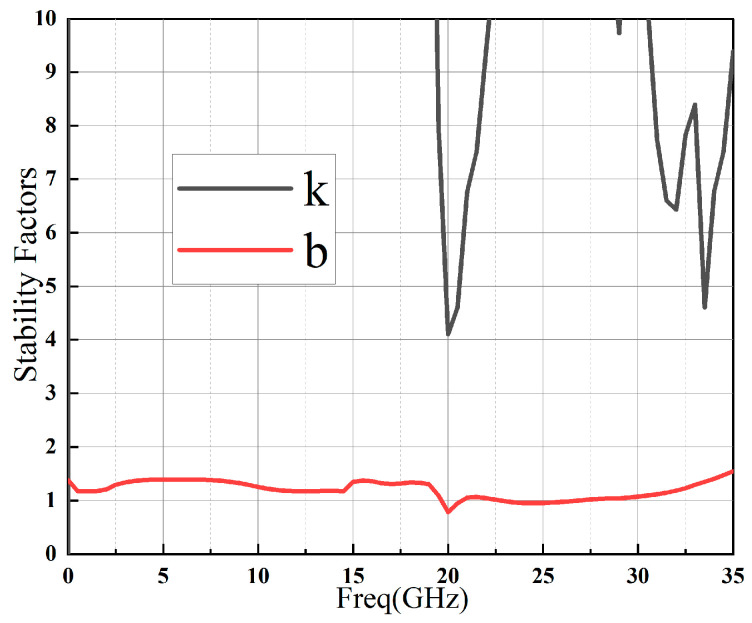
Measurement results of stability factors (k and b) for the fabricated MMIC LNA.

**Table 1 sensors-23-06611-t001:** Performances comparison with previously published works.

Ref.	Process	Freq (GHz)	Gain (dB)	NF (dB)	Area (mm^2^)	Power (mW)	Stages
[20]	CMOS	15.8–30.3	8.5–10.2	3.3–5.7	0.18	12.4	2
[21]	CMOS	21.2–24	16.5–19.1	3.6–4.2	0.35	0.99	2
[22]	CMOS	26–31	9.6–12	5.0–5.45	0.19	14.4	3
[23]	CMOS	21–34.5	10.5	2.4–3.9	0.03	4	2
[24]	GaAs	2–42	12.6–14.1	2.4	1.53	129	11
[25]	GaN-on-Si	35–37	19.7	3.7	5.21	/	4
[26]	GaN-on-Si	18–22.5	16.5–17.9	2.0–2.3	2.2	1100	2
[27]	GaN-on-SiC	27–31	12	3.7–3.9	4.08	560	4
[28]	GaN-on-SiC	27.5–28.5	18	4	6	6000	4
[29]	GaN-on-SiC	1–25	10–13	3.3–5	1.44	900	2
This Work	GaN-on-SiC	24.25–29.5	18.3–20.2	2.5–3.1	3.75	500	3

## Data Availability

The data may be obtained from the authors upon request.

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
