# Peer review of "An MMIC LNA for Millimeter-Wave Radar and 5G Applications with GaN-on-SiC Technology"

_sensors, 2023, doi:10.3390/s23146611_

Round 1

Reviewer 1 Report

This paper reports a good LNA design for FR2 5G and mmWave radar systems.

Please consider the following comments in the revised manuscript.

1. Introduction is too lengthy. I did not see the necessity. On the other hand, the novelty is not clearly shown. Focus on the tech instead of too much background info.

2. The extra stability circuit at the input bias will degrade the NF. Please analyze its effect on NF. Also, one of the advantages of GaN is its high power handling capability. I do not see any need to put an extra limiter for a GaN LNA. I have designed and measured GaN LNA to reach zero dB Gain (14 dB compression). And it is still alive with minor performance degradation.

3. Please cite and compare more recent references on TMTT and MWCL to show your design advantage.

4. The authors attribute the gap between measurement and simulation to parasitic effects which I could not agree with. Please be specific. What kind of parasitic effects had not been captured in WIN PDK?

Please try to use simple words and shorter sentences that you can handle well.

Author Response

Response to Reviewer 1 Comments

Point 1: Introduction is too lengthy. I did not see the necessity. On the other hand, the novelty is not clearly shown. Focus on the tech instead of too much background info.

Response 1: Thank you for your kind suggestion. The introduction section have been revised. More details about physical characteristics and microwave performance of the used GaN HEMT devices have been provided.

Point 2: The extra stability circuit at the input bias will degrade the NF. Please analyze its effect on NF. Also, one of the advantages of GaN is its high power handling capability. I do not see any need to put an extra limiter for a GaN LNA. I have designed and measured GaN LNA to reach zero dB Gain (14 dB compression). And it is still alive with minor performance degradation.

Response 2: Thank you for your question. The stability improvement at the input bias point is achieved by the absorption of the Miller capacitance through the RF-choke. The composition of the RC series network consists of a 50-ohm resistor in series with a large capacitor of 3.3pF, which ensures stability of the amplifier at low frequencies. The impact on noise is mainly at low frequencies, while its effect within the operating frequency range of this design is minimal.

As for the second point, this design does not use any additional limiters. Furthermore, in the introduction section, it is explained that one of the major advantages of GaN is its high power handling capability. The "limiter" mentioned by the reviewer may refer to the content outside the red box in the chip photo, which is unrelated to this design.

Point 3: Please cite and compare more recent references on TMTT and MWCL to show your design advantage.

Response 3: Thank you for your suggestion. We have cite and compare more recent references on TMTT and MWCL.

Point 4: The authors attribute the gap between measurement and simulation to parasitic effects which I could not agree with. Please be specific. What kind of parasitic effects had not been captured in WIN PDK?

Response 4: Generally, there would be gap between measurement and simulation when design a LNA. The design of the LNA in this paper used the same process as I did, and there is a difference of nearly 1 dB between the simulated NF and the measured NF (e.g., “25–31 GHz GaN-Based LNA MMIC Employing Hybrid-Matching Topology for 5G Base Station Applications”MWTL 2013 ). In addition, the WIN PDK used in this MMIC LNA is V1.0, and it has not undergone many iterations or updates.

However, what I meant was not that the PDK failed to capture certain parasitic effects, but rather that we didn't consider some external parasitic effects during the design phase. For example, when using a probe station, the probe itself introduces an inductance, and the inductance value of the wire bonding for supplying power to the PAD varies with the length of the wire. When simulating, an ideal ground is used, but this is not possible in the actual fabrication process, so it will also introduce a inductance. These factors contribute to gap between simulation and measurement results. Although these aspects have been included in the simulation, there will still inevitably be gap.

Reviewer 2 Report

The paper is accepted in the current format 

English Quality may improve a bit

Reviewer 3 Report

According to the Reviewer’s opinion, this study focuses on a relevant topic and presents interesting features. However, significant revisions and clarifications are required to make this manuscript worthy of publication in the journal Sensors.

In the Introduction the Authors should underline that a growing attention is being given to the GaN devices for low-noise applications, as they allow achieving respectable microwave noise performance, building HPA and LNAs in the same epitaxial material, and eliminating the receiver protection circuitry (reducing the overall RF front-end noise contribution), due to the excellent robustness of GaN transistors.

More details about physical characteristics and microwave performance of the used GaN HEMT devices have to be provided.

Please add labels to highlight the used GaN HEMTs in Fig. 8.

It would be interesting to report the frequency-dependent noise performance (e.g., noise parameters) of the used GaN HEMT devices.

The Authors should underline that a successful LNA design strongly depends on the extraction of an accurate noise transistor model and a couple of illustrative references might be provided on the GaN HEMT noise modeling (e.g., “GaN HEMT noise modeling based on 50-ohm noise factor,” MOTL 2015).

Please explain the reason of omitting the feedback resistance Rgd and the transconductance delay tau in the circuit topology in Fig. 3.

Please explain why you said that the “small‐signal equivalent circuit” in Fig. 3, which is supposed to be linear, is nonlinear.

The Authors may try to reduce the discrepancies between simulated and measured noise performance in Fig. 2 (e.g., by considering, in some way, the unaccounted parasitic effects during the simulation process).

The manuscript should be checked for grammar and typos (e.g., “. t is customary” should be replaced with “. It is customary”).

none

Round 2

Reviewer 3 Report

No further comments

None